# SGEM: STOCHASTIC GRADIENT WITH ENERGY AND MOMENTUM

## ABSTRACT

In this paper, we propose SGEM, Stochastic Gradient with Energy and Momentum to solve a large class of general non-convex stochastic optimization problems, based on the AEGD method that originated in the work [AEGD: Adaptive Gradient Descent with Energy. arXiv: 2010.05109]. SGEM incorporates both energy and momentum at the same time so as to inherit their dual advantages. We show that SGEM features an unconditional energy stability property, and derive energy-dependent convergence rates in the general nonconvex stochastic setting, as well as a regret bound in the online convex setting. A lower threshold for the energy variable is also provided. Our experimental results show that SGEM converges faster than AEGD and generalizes better or at least as well as SGDM in training some deep neural networks.

## 1 INTRODUCTION

In this paper, we propose SGEM: Stochastic Gradient with Energy and Momentum to solve the following general non-convex stochastic optimization problem

$$\min_{\theta \in \mathbb{R}^d} f(\theta) := \mathbb{E}_\xi[f(\theta; \xi)], \tag{1}$$

where $\mathbb{E}_\xi[\cdot]$ denotes the expectation with respect to the random variable $\xi$. We assume that $f$ is differentiable and bounded from below, i.e., $f^* = \inf_{\theta \in \mathbb{R}^d} f(\theta) > -c$ for some $c > 0$.

Problem (1) arises in many statistical learning and deep learning models (LeCun et al., 2015; Goodfellow et al., 2016; Bottou et al., 2018). For such large scale problems, it would be too expensive to compute the full gradient $\nabla f(\theta)$. One approach to handle this difficulty is to use an unbiased estimator of $\nabla f(\theta)$. Denote the stochastic gradient at the $t$-th iteration as $g_t$, the iteration of Stochastic Gradient Descent (SGD) (Robbins & Monro, 1951) can be described as:

$$\theta_{t+1} = \theta_t - \eta_t g_t,$$

where $\eta_t$ is called the learning rate. Its convergence is known to be ensured if $\eta_t$ meets the sufficient condition:

$$\sum_{t=1}^\infty \eta_t = \infty, \quad \sum_{t=1}^\infty \eta_t^2 < \infty. \tag{2}$$

However, vanilla SGD suffers from slow convergence due to the variance of the stochastic gradient, which is one of the major bottlenecks for practical use of SGD (Bottou, 2012; Shapiro & Wardi, 1996). Its performance is also sensitive to the learning rate, which is tricky to tune via (2). Different techniques have been introduced to improve the convergence and robustness of SGD, such as variance reduction (Defazio et al., 2014; Lei et al., 2017; Johnson & Zhang, 2013; Osher et al., 2019), momentum acceleration (Allen-Zhu, 2018; Sutskever et al., 2013), and adaptive learning rate (Duchi et al., 2011; Tieleman & Hinton, 2012; Kingma & Ba, 2017). Among these, momentum and adaptive learning rate techniques are most economic since they require slightly more computation in each iteration. However, training with adaptive algorithms such as Adam or its variants typically generalizes worse than SGD with momentum (SGDM), even when the training performance is better Wilson et al. (2018).

The most popular momentum technique, Heavy Ball (HB) (Polyak, 1964) has been extensively studied for stochastic optimization problems (Liu et al., 2020b; Jin et al., 2018; Qian, 1999). SGDM,

also called SHB, as a combination of SGD and momentum takes the following form

$$m_t = \mu m_{t-1} + g_t, \ \theta_{t+1} = \theta_t - \eta_t m_t,$$

where $m_0 = 0$ and $\mu \in (0, 1)$ is the momentum factor. This helps to reduce the variance in stochastic gradients thus speeds up the convergence, and has been found to be successful in practice (Sutskever et al., 2013).

AEGD originated in the work Liu & Tian (2020) is a gradient-based optimization algorithm that adjusts the learning rate by a transformed gradient $v$ and an energy variable $r$. The method includes two ingredients: the base update rule:

$$\theta_{t+1} = \theta_t + 2\eta r_{t+1} v_t, \quad r_{t+1} = \frac{r_t}{1 + 2\eta v_t^2}, \tag{3}$$

and the stochastic evaluation of the transformed gradient $v_t$ as

$$v_t = \frac{g_t}{2\sqrt{f(\theta_t; \xi_t) + c}}. \tag{4}$$

AEGD is unconditionally energy stable with guaranteed convergence in energy regardless of the size of the base learning rate $\eta > 0$ and how $v_t$ is evaluated. This explains why the method can have a rapid initial training process as well as good generalization performance (Liu & Tian, 2020).

In this paper, we attempt to incorporate both energy and momentum at the same time so as to inherit their dual advantages. We do so by keeping the base AEGD update rule (3), but taking

$$v_t = \frac{m_t}{2(1 - \beta^t)\sqrt{f(\theta_t; \xi_t) + c}}, \quad m_t = \beta m_{t-1} + (1 - \beta)g_t, \quad \beta \in (0, 1). \tag{5}$$

We call this novel method SGEM. An immediate advantage is that with such $v_t$ one can significantly reduce the oscillations observed in the AEGD in stochastic cases. Regarding the theoretical results, in this work we develop a convergence theory for SGEM, in both stochastic nonconvex setting and online convex setting. While in Liu & Tian (2020), convergence analysis is provided mainly in deterministic setting, and the result in the stochastic setting is only an upper bound on the norm of the stochastic transformed gradient $v$ rather than on $\nabla f(\theta)$.

We highlight the main contributions of our work as follows:

- We propose a novel and simple gradient-based method SGEM which integrates both energy and momentum. The only hyperparameter requires tuning is the base learning rate.

- We show the unconditional energy stability of SGEM, and provide energy-dependent convergence rates in the general stochastic nonconvex setting, and a regret bound for the online convex framework. We also obtain a lower threshold for the energy variable. Our assumptions are natural and mild.

- We empirically validate the good performance of SGEM on several deep learning benchmarks. Our results show that

  - The base learning rate requires little tuning on complex deep learning tasks.
  - Overall, SGEM is able to achieve both fast convergence and good generalization performance. Specifically, SGEM converges faster than AEGD and generalizes better or at least as well as SGDM.

**Related works.** The essential idea behind AEGD is the so called Invariant Energy Quadratizaton (IEQ) strategy, originally introduced for developing linear and unconditionally energy stable schemes for gradient flows in the form of partial differential equations (Yang, 2016; Zhao et al., 2017). As for gradient-based methods, there has appeared numerous works on the analysis of convergence rates. In online convex setting, a regret bound for SGD is derived in Zinkevich (2003); the classical convergence results of SGD in stochastic nonconvex setting can be found in Bottou et al. (2018); For SGDM, we refer the readers to Yu et al. (2019); Yan et al. (2018); Liu et al. (2020b) for convergence rates on smooth nonconvex objectives. For adaptive gradient methods, most convergence analysis are restricted to online convex setting (Duchi et al., 2011; Reddi et al., 2018; Luo et al., 2019), while recent attempts, such as Chen et al. (2019); Zou et al. (2019), have been made to analyze the convergence in stochastic nonconvex setting.

This paper is organized as follows. We first review AEGD in Section 2, then introduce the proposed algorithm in Section 3. Theoretical analysis including unconditional energy stability, convergence rates in both stochastic nonconvex setting and online convex setting are presented in Section 4. In Section 5, we report some experimental results on deep learning tasks.

**Notation** For a vector $\theta \in \mathbb{R}^n$, we denote $\theta_{t,i}$ as the $i$-th element of $\theta$ at the $t$-th iteration. For vector norm, we use $\|\cdot\|$ to denote $l_2$ norm and use $\|\cdot\|_\infty$ to denote $l_\infty$ norm. We also use $[m]$ to represent the list $\{1, ..., m\}$ for any positive integer $m$.

## 2 REVIEW OF AEGD

Recall that for the objective function $f$, we assume that $f$ is differentiable and bounded from below, i.e., $f(\theta) > -c$ for some $c > 0$. The key idea of AEGD introduced in Liu & Tian (2020) is the use of an auxiliary energy variable $r$ such that

$$\nabla f(\theta) = 2rv, \quad v := \nabla\sqrt{f(\theta) + c}, \tag{6}$$

where $r$, taking as $\sqrt{f(\theta) + c}$ initially, will be updated together with $\theta$, and $v$ is dubbed as the transformed gradient. The gradient flow $\dot{\theta} = -\nabla f(\theta)$ is then replaced by

$$\dot{\theta} = -2rv, \quad \dot{r} = v \cdot \dot{\theta}.$$

A simple implicit-explicit discretization gives the following AEGD update rule:

$$v_t = \frac{\nabla f(\theta_t)}{2\sqrt{f(\theta_t) + c}}. \tag{7a}$$

$$\theta_{t+1} = \theta_t - 2\eta r_{t+1} v_t, \tag{7b}$$

$$r_{t+1} - r_t = v_t \cdot (\theta_{t+1} - \theta_t). \tag{7c}$$

This yields a decoupled update for $r$ as $r_{t+1} = r_t/(1 + 2\eta|v_t|^2)$, which serves to adapt the learning rate. For large-scale problems, stochastic sampling approach is preferred. Let $f(\theta_t; \xi_t)$ be a stochastic estimator of the function value $f(\theta_t)$ at the $t$-th iteration, $g_t$ be a stochastic estimator of the gradient $\nabla f(\theta_t)$, then the stochastic version of AEGD is still (7) but with $v_t$ replaced by

$$v_t = \frac{g_t}{2\sqrt{f(\theta_t; \xi_t) + c}}.$$

Usually, $g_t$ should be required to satisfy $\mathbb{E}[g_t] = \nabla f(\theta_t)$ and $\mathbb{E}[\|g_t\|^2]$ bounded. Correspondingly, an element-wise version of AEGD for stochastic training reads as

$$v_{t,i} = \frac{g_{t,i}}{2\sqrt{f(\theta_t; \xi_t) + c}}, \quad i \in [n], \tag{8a}$$

$$r_{t+1,i} = \frac{r_{t,i}}{1 + 2\eta v_{t,i}^2}, \quad r_{1,i} = \sqrt{f(\theta_1; \xi_1) + c}, \tag{8b}$$

$$\theta_{t+1,i} = \theta_{t,i} - 2\eta r_{t+1,i} v_{t,i}. \tag{8c}$$

The element-wise AEGD allows for different effective learning rates for different coordinates, which has been empirically verified to be more effective than the global AEGD (7). For further details, we refer to Liu & Tian (2020). We will focus only on the element-wise version of SGEM in what follows.

## 3 THE PROPOSED ALGORITHM

In this section, we present a novel algorithm to improve AEGD with added momentum in the following manner:

$$m_t = \beta m_{t-1} + (1 - \beta)g_t, \quad m_0 = \mathbf{0}, \tag{9a}$$

$$v_t = \frac{m_t}{2(1 - \beta^t)\sqrt{f(\theta_t; \xi_t) + c}}, \tag{9b}$$

where $\beta \in (0,1)$ controls the weight for gradient at each step. With $v_t$ so defined, the update rule for $r$ and $\theta$ are kept the same as given in (8b, c). The relation between the energy and the momentum in the algorithm is realized through relating $m_t$ ( as an approximation to $\nabla f$) to $v_t$ (as an approximation of $\nabla F = \frac{\nabla f}{2\sqrt{f+c}}$), where $v_t$ is used to update the energy $r_{t+1}$. In machine learning tasks, $f$ as a loss function is often in the form of $f(\theta) = \frac{1}{m}\sum_{i=1}^{m} l_i(\theta)$, where $l_i$, measuring the distance between the model output and target label at the $i$-th data point, is typically bounded from below, that is, $l_i(\theta) > -c, \forall i \in [m]$, for some $c > 0$. Hence $c$ in (9b) can be easily chosen in advance so that $f(\theta_t; \xi_t)$ as a random sample from $\{l_i(\theta_t)\}_{i=1}^{m}$ is bounded below by $-c$ for all $t \in [T]$. We summarize this in Algorithm 1 (called SGEM, for short).

A key feature of SGEM is that it incorporates momentum into AEGD without changing the overall structure of the AEGD algorithm (the update of $r$ and $\theta$ remain the same) so that it is shown (in Section 4) to still enjoy the unconditional energy stability property as AEGD does. In addition, by using $m_t$ instead of $g_t$, the variance can be largely reduced. In fact, as proved in Liu et al. (2020b), under the assumption $\mathbb{E}_{\xi_t}[\|g_t - \nabla f(\theta_t)\|^2] = \sigma_g^2 < \infty$, $m_t$, which can be expressed as a linear combination of the gradients at all previous steps,

$$m_t = (1-\beta)\sum_{j=1}^{t} \beta^{t-j} g_j, \tag{10}$$

enjoys a reduced "variance" in the sense that

$$\mathbb{E}_{\xi_t}\left[\left\|m_t - (1-\beta)\sum_{j=1}^{t}\beta^{t-j}\nabla f(\theta_j)\right\|^2\right] \le (1-\beta)\sigma_g^2.$$

---

**Algorithm 1** SGEM. Good default setting for parameters are $\eta = 0.2$, $\beta = 0.9$

---

**Require:** the base learning rate $\eta$; a constant $c$ such that $f(\theta_t; \xi_t) + c > 0$ for all $t \in [T]$; a momentum factor $\beta \in (0,1)$.
**Require:** Initialization: $\theta_1$; $m_0 = \mathbf{0}$; $r_1 = \sqrt{f(\theta_1; \xi_1) + c}\,\mathbf{1}$
 1: **for** $t = 1$ to $T-1$ **do**
 2:    Compute gradient: $g_t = \nabla f(\theta_t; \xi_t)$
 3:    $m_t = \beta m_{t-1} + (1-\beta)g_t$ (momentum update)
 4:    $v_t = m_t/(2(1-\beta^t)\sqrt{f(\theta_t; \xi_t) + c})$ (transformed momentum)
 5:    $r_{t+1} = r_t/(1 + 2\eta v_t \odot v_t)$ (energy update)
 6:    $\theta_{t+1} = \theta_t - 2\eta r_{t+1} \odot v_t$ (state update)
 7: **end for**
 8: **return** $\theta_T$

---

**Remark 3.1.** *(i) In Algorithm 1, we use $x \odot y$ to denote element-wise product, $x/y$ to denote element-wise division of two vectors $x, y \in \mathbb{R}^n$.*
*(ii) It is clear that $m_t$ defined in (10) is not a convex combination of $g_j$, this is why there is a factor $1 - \beta^t$ in (9b); such treatment is dubbed as bias correction in Kingma & Ba (2017) for Adam.*
*(iii) In most machine learning problems, we have $f(\theta) \ge 0$, for which a good default value for $c$ in Algorithm 1 is $1$.*

## 4 THEORETICAL RESULTS

In this section, we present our theoretical results, including the unconditional energy stability of SGEM, the convergence of SGEM for the general stochastic nonconvex optimization, a lower bound for energy $r_T$, and a regret bound in the online convex setting.

### 4.1 UNCONDITIONAL ENERGY STABILITY

**Theorem 4.1.** *(Unconditional energy stability) SGEM in Algorithm 1 is unconditionally energy stable in the sense that for any step size $\eta > 0$,*

$$\mathbb{E}[r_{t+1,i}^2] = \mathbb{E}[r_{t,i}^2] - \mathbb{E}[(r_{t+1,i} - r_{t,i})^2] - \eta^{-1}\mathbb{E}[(\theta_{t+1,i} - \theta_{t,i})^2], \quad i \in [n], \tag{11}$$

*that is $\mathbb{E}[r_{t,i}]$ is strictly decreasing and convergent with $\mathbb{E}[r_{t,i}] \to \mathbb{E}[r_i^*]$ as $t \to \infty$, and also*

$$\lim_{t \to \infty} \mathbb{E}[(\theta_{t+1,i} - \theta_{t,i})^2] = 0, \quad \sum_{t=1}^{\infty} \mathbb{E}[(\theta_{t+1,i} - \theta_{t,i})^2] \le \eta(f(\theta_1) + c), \quad \forall i \in [n]. \qquad (12)$$

**Remark 4.1.** *(i) The unconditional energy stability only depends on (8b, c), irrespective of the choice for $v_t$. This property essentially means that the energy variable $r_t$, which serves to approximate $\sqrt{f(\theta_t) + c}$, is strictly decreasing for any $\eta > 0$.*
*(ii) (12) indicates that the sequence $\|\theta_{t+1} - \theta_t\|$ converges to zero at a rate of at least $1/\sqrt{t}$. We note that this does not guarantee the convergence of $\{\theta_t\}$ unless additional information on the geometry of $f$ is available.*

*Proof.* From (8b, c) we have

$$\begin{aligned}
(\theta_{t+1,i} - \theta_{t,i})^2 &= 4\eta^2 r_{t+1,i}^2 v_{t,i}^2 \quad \text{(By 8c)} \\
&= (2\eta r_{t+1,i})(r_{t,i} - r_{t+1,i}) \quad \text{(By 8b)} \\
&= \eta((r_{t,i}^2 - r_{t+1,i}^2) - (r_{t,i} - r_{t+1,i})^2).
\end{aligned}$$

This upon taking expectation ensures the asserted properties. Such proof with no use of the special form of $v_t$, is the same as that for AEGD (see Liu & Tian (2020)). $\qquad \square$

## 4.2 CONVERGENCE ANALYSIS

Below, we state the necessary assumptions that are commonly used for analyzing the convergence of a stochastic algorithm for nonconvex problems, and notations that will be used in our analysis.

**Assumption 4.1.** *1. (**Smoothness**) The objective function in (1) is $L$-smooth: for any $x, y \in \mathbb{R}^n$,*

$$f(y) \le f(x) + \nabla f(x)^\top (y - x) + \frac{L}{2} \|y - x\|^2.$$

*2. (**Independent samples**) The random samples $\{\xi_t\}_{t=1}^{\infty}$ are independent.*

*3. (**Unbiasedness**) The estimator of the gradient and function value are unbiased:*

$$\mathbb{E}_{\xi_t}[g_t] = \nabla f(\theta_t), \quad \mathbb{E}_{\xi_t}[f(\theta_t; \xi_t)] = f(\theta_t).$$

Denoting the variance of the stochastic gradient and function value by $\sigma_g$ and $\sigma_f$, respectively:

$$\mathbb{E}_{\xi_t}[\|g_t - \nabla f(\theta_t)\|^2] = \sigma_g^2, \quad \mathbb{E}_{\xi_t}[|f(\theta_t; \xi_t) - f(\theta_t)|^2] = \sigma_f^2.$$

We have the following results.

**Theorem 4.2.** *Let $\{\theta_t\}$ be the solution sequence generated by Algorithm 1 with a fixed $\eta > 0$. Under Assumption 4.1 and assume that the stochastic gradient and function value are bounded such that $\|g_t\|_\infty \le G_\infty$ and $0 < a \le f(\theta_t; \xi_t) + c \le B$, then $\sigma_g \le G_\infty$ and for all $T \ge 1$,*

$$\frac{1}{T} \mathbb{E}\left[ \min_i r_{T,i} \sum_{t=1}^{T} \|\nabla f(\theta_t)\|^2 \right] \le \frac{C_1 + C_2 n + C_3 \sigma_g \sqrt{nT}}{\eta T},$$

*where $C_1, C_2, C_3$ are constants depending on $\beta, \eta, L, G_\infty, a, B, n$ and $f(\theta_1) + c$.*

**Remark 4.2.** *(i) Numerically we observe that for reasonable choice of $\eta$, $r_{t,i}$ decays much slower than $1/\sqrt{t}$ (See Figures 1), thus the convergence result in Theorem 4.2 is meaningful. The question of how $r_T$ depends on $T$ is theoretically interesting but subtle to characterize. Nevertheless, in Theorem 4.3 below, we identify a sufficient condition for ensuring a lower threshold for $\mathbb{E}[r_{T,i}]$, from which we see that in the absence of noise, i.e $\sigma_g = 0$, $\min_i r_i^* > 0$ can be ensured, then the rate of $O(1/T)$ is recovered in Theorem 4.2.*
*(ii) The assumption that the magnitude of the stochastic gradient is bounded is standard in nonconvex stochastic analysis (Bottou et al., 2018). As for the upper bound on the stochastic function value, we recall the new introduced update rule in SGEM (9a,b): to bound $v_t$, we don't need an upper bound on $f$; while such upper bound is technically needed to bound $m_t$ since $m_t = 2(1 - \beta^t)\sqrt{f + c} v_t$.*

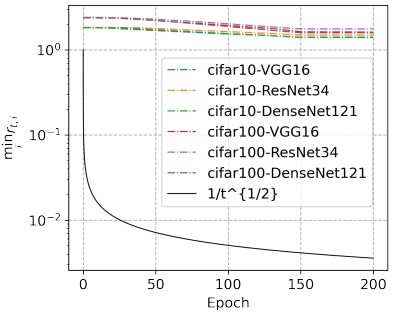

Figure 1: $\min_i r_{t,i}$ of SGEM with default base learning rate 0.2 in training DL tasks.

We only present a sketch of proofs for Theorem 4.2 and 4.3 here, using notation $\tilde{F}_t = \sqrt{f(\theta_t; \xi_t) + c}$, $\eta_t = \eta/\tilde{F}_t$ and viewing $r_{t+1}$ as a $n \times n$ diagonal matrix that is made up of $[r_{t+1,1}, ..., r_{t+1,i}, ..., r_{t+1,n}]$. Detailed proofs, including two crucial lemmas and the full proof for Theorem 4.4, are deferred to the appendix.

*Proof.* Using the $L$-smoothness of $f$, we have

$$f(\theta_{t+1}) - f(\theta_t) \leq \nabla f(\theta_t)^\top (\theta_{t+1} - \theta_t) + \frac{L}{2} \|\theta_{t+1} - \theta_t\|^2. \tag{13}$$

The first term on the RHS is carefully regrouped as

$$-\frac{1-\beta}{1-\beta^t} \nabla f(\theta_t)^\top \eta_{t-1} r_t g_t + \frac{1-\beta}{1-\beta^t} \nabla f(\theta_t)^\top (\eta_{t-1} r_t - \eta_t r_{t+1}) g_t - \frac{\beta}{1-\beta^t} \nabla f(\theta_t)^\top \eta_t r_{t+1} m_{t-1}.$$

Taking a conditional expectation on the first term gives

$$\frac{1-\beta}{1-\beta^t} \eta_{t-1} \nabla f(\theta_t)^T r_t \nabla f(\theta_t) \geq (1-\beta) \frac{\eta}{\sqrt{B}} \min_i r_{t,i} \|\nabla f(\theta_t)\|^2.$$

We manage to bound the other two terms in terms of $\sum_{i=1}^n \sum_{t=1}^T r_{t+1,i} g_{t,i}^2$ and $\sum_{i=1}^n \sum_{t=1}^T r_{t+1,i} m_{t,i}^2$. Their bounds are presented in Lemma A.2. The asserted bound then follows by further summation in $t$ with telescope cancellation for $f(\theta_{t+1}) - f(\theta_t)$ and bounding the last term in (13) using (12). □

### 4.3 LOWER BOUND FOR THE ENERGY

First note that the $L$-smoothness of $f(\theta)$ implies the $L_F$-smoothness of $F(\theta) = \sqrt{f(\theta) + c}$ with

$$L_F = \frac{1}{2F(\theta^*)} \left( L + \frac{G_\infty^2}{2F^2(\theta^*)} \right). \tag{14}$$

This will be used in the following result and its proof.

**Theorem 4.3** (Lower bound of $r_T$). *Under the same assumptions as in Theorem 4.2, we have*

$$\min_i \mathbb{E}[r_{T,i}] \geq \max\{F(\theta^*) - \eta D_1 - \beta D_2 - \sigma D_3, 0\}, \tag{15}$$

*where $\sigma = \max\{\sigma_f, \sigma_g\} \leq \max\{G_\infty, B\}$ with $L_F$ given in (14) and*

$$D_1 = \frac{L_F n F^2(\theta_1)}{2}, \quad D_2 = \frac{\sqrt{B} n F(\theta_1)}{(1-\beta)\sqrt{a}},$$

$$D_3 = \frac{1}{2\sqrt{a}} + F(\theta_1)\sqrt{\eta n T}\sqrt{\frac{G_\infty^2}{4a^3} + \frac{1}{a}}.$$

*Moreover, in the absence of noise, we have*

$$\min_i r_{T,i} > \min_i r_i^* > 0 \quad if \quad \eta D_1 + \mu D_2 < F(\theta^*). \tag{16}$$

**Remark 4.3.** *(i) (16) is only a sufficient condition, not used as a guide for choosing $\eta$. We observe from our experimental results that the upper bound for $\eta$ to guarantee the positiveness of $r_i^*$ can be much larger (See Figure 1).*
*(ii) In Theorem 4.3, we measure how far $r^*$ can deviate from $F(\theta^*)$ in the worst situation. Under the stochastic nonconvex setting, $\eta D_1$ is the error brought by the step size $\eta$, $\beta D_2$ is due to the use of momentum, and $\sigma D_3$ is responsible for the existence of noise.*
*(iii) In the case of no momentum and no noise, we have*

$$\min_i r_{T,i} > \min_i r_i^* > 0 \quad if \quad \eta < \frac{F(\theta^*)}{D_1}.$$

*This captures the result for the deterministic AEGD obtained in Liu & Tian (2020).*

*Proof.* Using the $L_F$-smoothness of $F(\theta)$, we have

$$F(\theta_{t+1}) - F(\theta_t) \leq \nabla F(\theta_t)^\top (\theta_{t+1} - \theta_t) + \frac{L_F}{2}\|\theta_{t+1} - \theta_t\|^2, \tag{17}$$

in which the key term $\nabla F(\theta_t)^\top (\theta_{t+1} - \theta_t)$ can be decomposed into three terms:

$$(\nabla F(\theta_t) - \frac{g_t}{2\tilde{F}_t})^\top (\theta_{t+1} - \theta_t), \quad (\frac{g_t}{2\tilde{F}_t} - \frac{v_t}{1-\beta})^\top (\theta_{t+1} - \theta_t), \quad (\frac{v_t}{1-\beta})^\top (\theta_{t+1} - \theta_t),$$

The first two terms are bounded by using the bounded variance assumption and (12), respectively. We convert the last term, using (recall 7c)

$$r_{t+1,i} - r_{t,i} = v_{t,i}(\theta_{t+1,i} - \theta_{t,i}),$$

into expressions in terms of $r_{t+1,i} - r_{t,i}$, which upon summation is bounded by $r_{T,i}$. The last term in (17) is bounded again by using (12). □

## 4.4 REGRET BOUND FOR ONLINE CONVEX OPTIMIZATION

Our algorithm is also applicable to the online optimization that deals with the optimization problems having no or incomplete knowledge of the future (online). In the framework proposed in Zinkevich (2003), at each step $t$, the goal is to predict the parameter $\theta_t \in \mathcal{F}$, where $\mathcal{F} \subset \mathbb{R}^n$ is a feasible set, and evaluate it on a previously unknown loss function $f_t$. The nature of the sequence is unknown in advance, the SGEM algorithm needs to be modified. This can be done by replacing $f(\theta_t, \xi_t)$ by $f_t(\theta_t)$ and taking $g_t = \nabla f_t(\theta_t)$ in $v_t$ defined in (9), i.e.,

$$m_t = \beta m_{t-1} + (1-\beta)\nabla f_t(\theta_t), \tag{18a}$$

$$v_t = \frac{m_t}{2(1-\beta^t)\sqrt{f_t(\theta_t) + c}}. \tag{18b}$$

This algorithm is also unconditional energy stable as pointed out in Remark 4.1. For convergence, we evaluate our algorithm using the regret, that is the sum of all the previous difference between the online prediction $f_t(\theta_t)$ and the best fixed point parameter $f_t(\theta^*)$ from a feasible set $\mathcal{F}$:

$$R(T) = \sum_{t=1}^{T}[f_t(\theta_t) - f_t(\theta^*)],$$

where $\theta^* = \text{argmin}_{\theta \in \mathcal{F}} \sum_{t=1}^{T} f_t(\theta)$. For convex objectives we have the following regret bound.

**Theorem 4.4.** *Let $\{\theta_t\}$ be the solution sequence generated by SGEM with a fixed $\eta > 0$. Assume that $\|x - y\|_\infty \leq D_\infty$ for all $x, y \in \mathcal{F}$, $0 < a \leq f_t(\theta_t) + c \leq B$, and $\theta_t \in \mathcal{F}$ for all $t \in [T]$. When $\mathcal{F}$ and $f_t$ are convex, SGEM achieves the following bound on the regret, for all $T \geq 1$,*

$$R(T) \leq C\sqrt{nT/\eta} \left(\sum_{i=1}^{n} \frac{1}{r_{T,i}}\right)^{1/2}, \tag{19}$$

*where $C$ is a constant depending on $\beta, B, D_\infty$ and $f_1(\theta_1) + c$.*

**Remark 4.4.** *(i) If $r_{T,i} > r_i^* > 0$, then $R(T)$ is of order $O(\sqrt{T})$, which is known the best possible bound for online convex optimization (Hazan, 2019, Section 3.2). Our experimental results show that for $\eta$ in a reasonable range, $r_{T,i}$ decays much slower than $1/\sqrt{T}$ (See Figure 1), for which the convergence holds true in the sense that*

$$\lim_{T \to \infty} \frac{R(T)}{T} = 0.$$

*(ii) The bound on $\theta_t$ is typically enforced by projection onto $\mathcal{F}$ (Zinkevich, 2003), with which the regret bound (19) can still be proven since projection is a contraction operator (Hazan, 2019, Chapter 3). As for the upper bound on the function value, just like we remarked for Theorem 4.2, it is technically needed to bound $m_t$.*

## 5 NUMERICAL EXPERIMENTS

In this section, we compare the performance of the proposed method with several other methods, including AEGD, SGDM, AdaBelief (Zhuang et al., 2020), AdaBound (Luo et al., 2019), RAdam (Liu et al., 2020a), Yogi (Zaheer et al., 2018), and Adam (Kingma & Ba, 2017), when applied to training deep neural networks. [1] We consider three convolutional neural network (CNN) architectures: VGG-16 (Simonyan & Zisserman, 2015), ResNet-34 (He et al., 2016), DenseNet-121 (Huang et al., 2017) on the CIFAR-10 and CIFAR-100 datasets (Krizhevsky & Hinton, 2009); we also conduct experiments on the ImageNet dataset (Russakovsky et al., 2015) with the ResNet-18 architecture (He et al., 2016).

For experiments on CIFAR-10 and CIFAR-100, we employ the fixed budget of 200 epochs and reduce the learning rates by 10 after 150 epochs. The weight decay and minibatch size are set as $5 \times 10^{-4}$ and 128 respectively. For the ImageNet tasks, we run 90 epochs and use similar learning rate decaying strategy at the 30th and 60th epoch. The weight decay and minibatch size are set as $1 \times 10^{-4}$ and 256 respectively.

In each task, we only tune the base learning rate and report the one that achieves the best final generalization performance for each method:

- SGEM: For CIFAR10 & 100 tasks, we use the default parameter $\eta = 0.2$; for the ImageNet task, the learning rate is set as $\eta = 0.3$.
- SGDM, AEGD: We search learning rate among $\{0.05, 0.1, 0.2\}$.
- AdaBelief, AdaBound, Yogi, RAdam, Adam: We search learning rate among $\{0.0005, 0.001, 0.01\}$, other hyperparameters such as $\beta_1, \beta_2, \epsilon$ are set as the default values in their literature.

From the experimental results of CIFAR10 & 100, we see that in all tasks, SGEM and AEGD achieve higher test accuracy than the other methods while the oscillation of AEGD in test accuracy is significantly reduced by SGEM as expected. We also observe that the differences between these methods are more obvious in experiments on CIFAR-100.

For the ImageNet task, since all previous experiments show that SGDM gives the highest test accuracy, we focus on the comparison between SGDM and SGEM, and only run Adam as a representative of other adaptive methods. The results are presented in Figure 3. It can be seen that SGEM still shows fast convergence and is able to achieve comparable test accuracy as SGDM in the end of training. Here the highest test accuracy achieved by SGDM and SGEM are 69.89 and 69.92, respectively.

## 6 CONCLUSION

In this paper, we propose SGEM, which integrates AEGD with momentum. We show that SGEM still enjoys the unconditional energy stability property as AEGD, while the use of momentum helps to reduce the variance of the stochastic gradient significantly, as verified in our experiments. We

---

[1]Code is available at https://anonymous.4open.science/r/SGDEM-0042.

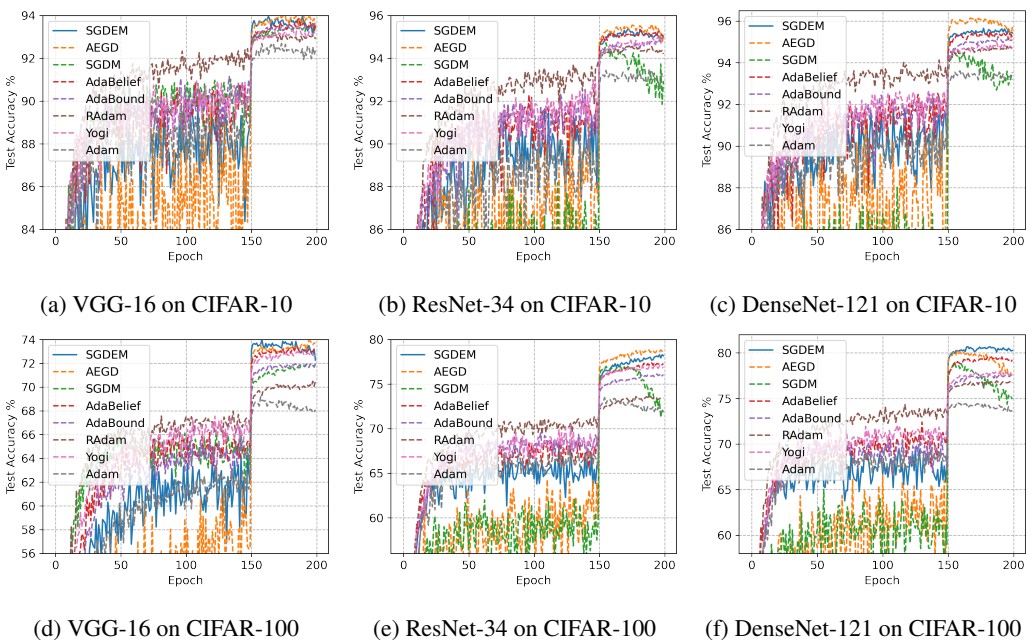

(a) VGG-16 on CIFAR-10     (b) ResNet-34 on CIFAR-10     (c) DenseNet-121 on CIFAR-10

(d) VGG-16 on CIFAR-100     (e) ResNet-34 on CIFAR-100     (f) DenseNet-121 on CIFAR-100

Figure 2: Test accuracy for VGG-16, ResNet-34 and DenseNet-121 on CIFAR-10/100

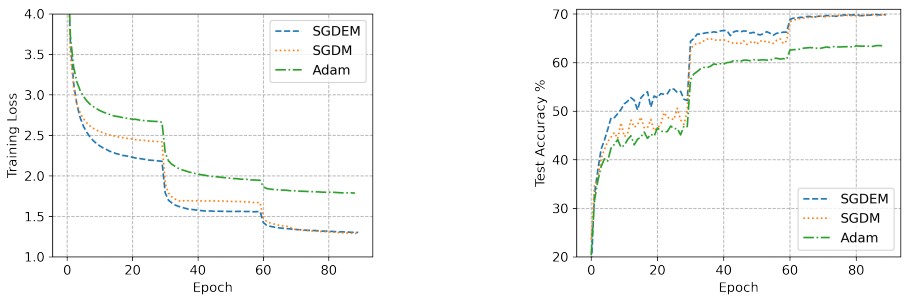

Figure 3: Training loss and test accuracy for ResNet-18 on ImageNet

also provide convergence analysis in both online convex setting and the general stochastic nonconvex setting. Since our convergence results depend on the energy variable, a lower bound on the energy is also presented. Finally, we empirically show that SGEM converges faster than AEGD and generalizes better or at least as well as SGDM on several deep learning benchmarks.

Based on our observations in this paper, we list some problems for future work. First, we believe there is a threshold for $\eta^*$, such that $r_T$ either tends to a positive number or decays slower than $1/\sqrt{T}$ if $\eta < \eta^*$. A further theoretical investigation on this issue is desirable. Second, since $r_t$ is strictly decreasing, there is a room to limit $r_t$ for controlling its decay whenever necessary. A proper energy limiter should be obtained.

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

# A APPENDIX

## A.1 PROOF OF THEOREM 4.2

For the proofs of Theorem 4.2 and Theorem 4.3, we introduce notation

$$\tilde{F}_t := \sqrt{f(\theta_t; \xi_t) + c}. \tag{20}$$

The initial data for $r_i$ is taken as $r_{1,i} = \tilde{F}_1$. We also denote the update rule presented in Algorithm 1 as

$$\theta_{t+1} = \theta_t - 2\eta r_{t+1} v_t, \tag{21}$$

where $r_{t+1}$ is viewed as a $n \times n$ diagonal matrix that is made up of $[r_{t+1,1}, ..., r_{t+1,i}, ..., r_{t+1,n}]$.

**Lemma A.1.** *Under the assumptions in Theorem 4.2, we have for all $t \in [T]$,*

(i) $\|\nabla f(\theta_t)\|_\infty \leq G_\infty.$

(ii) $\mathbb{E}[(\tilde{F}_t)^2] = F^2(\theta_t) = f(\theta_t) + c.$

(iii) $\mathbb{E}[\tilde{F}_t] \leq F(\theta_t).$ *In particular, $\mathbb{E}[r_{1,i}] = \mathbb{E}[\tilde{F}_1] \leq F(\theta_1)$ for all $i \in [n]$.*

(iv) $\sigma_g^2 = \mathbb{E}[\|g_t - \nabla f(\theta_t)\|^2] \leq G_\infty^2$ *and* $\sigma_f^2 = \mathbb{E}[|f(\theta_t; \xi_t) - f(\theta_t)|^2] \leq B^2.$

(v) $\mathbb{E}[|F(\theta_t) - \tilde{F}_t|] \leq \frac{1}{2\sqrt{a}}\sigma_f.$

(vi) $\mathbb{E}[\|\nabla F(\theta_t) - \frac{g_t}{2\tilde{F}_t}\|^2] \leq \frac{G_\infty^2}{8a^3}\sigma_f^2 + \frac{1}{2a}\sigma_g^2.$

*Proof.* (i) By assumption $\|g_t\|_\infty \leq G_\infty$, we have

$$\|\nabla f(\theta_t)\|_\infty = \|\mathbb{E}[g_t]\|_\infty \leq \mathbb{E}[\|g_t\|_\infty] \leq G_\infty.$$

(ii) This follows from the unbiased sampling of

$$f(\theta_t) = \mathbb{E}_{\xi_t}[f(\theta_t; \xi_t)].$$

(iii) By Jensen's inequality, we have

$$\mathbb{E}[\tilde{F}_t] \leq \sqrt{\mathbb{E}[\tilde{F}_t^2]} = \sqrt{F(\theta_t)^2} = F(\theta_t).$$

(iv) By assumptions $\|g_t\|_\infty \leq G_\infty$ and $f(\theta_t; \xi_t) + c < B$, we have

$$\sigma_g^2 = \mathbb{E}[\|g_t - \nabla f(\theta_t)\|^2] = \mathbb{E}[\|g_t\|^2] - \|\nabla f(\theta_t)\|^2 \leq G_\infty^2,$$

$$\sigma_f^2 = \mathbb{E}[\|f(\theta_t; \xi_t) - f(\theta_t)\|^2] = \mathbb{E}[\|f(\theta_t; \xi_t)\|^2] - \|f(\theta_t)\|^2 \leq B^2.$$

(v) By the assumption $0 < a \leq f(\theta_t; \xi_t) + c = \tilde{F}_t^2$, we have

$$\mathbb{E}[|F(\theta_t) - \tilde{F}_t|] \leq \mathbb{E}\left[\left|\frac{f(\theta_t) - f(\theta_t; \xi_t)}{F(\theta_t) + \tilde{F}_t}\right|\right] \leq \frac{1}{2\sqrt{a}}\mathbb{E}[|f(\theta_t) - f(\theta_t; \xi_t)|] \leq \frac{1}{2\sqrt{a}}\sigma_f.$$

(vi) By the definition of $F(\theta)$, we have

$$\begin{aligned}
\|\nabla F(\theta_t) - \frac{g_t}{2\tilde{F}_t}\|^2 &= \left\|\frac{\nabla f(\theta_t)}{2F(\theta_t)} - \frac{g_t}{2\tilde{F}_t}\right\|^2 \\
&= \frac{1}{4}\left\|\frac{\nabla f(\theta_t)(\tilde{F}_t - F(\theta_t))}{F(\theta_t)\tilde{F}_t} + \frac{\nabla f(\theta_t) - g_t}{\tilde{F}_t}\right\|^2 \\
&\leq \frac{1}{2}\left\|\frac{\nabla f(\theta_t)(\tilde{F}_t - F(\theta_t))}{F(\theta_t)\tilde{F}_t}\right\|^2 + \frac{1}{2}\left\|\frac{\nabla f(\theta_t) - g_t}{\tilde{F}_t}\right\|^2 \\
&\leq \frac{G_\infty^2}{2a^2}|\tilde{F}_t - F(\theta_t)|^2 + \frac{1}{2a}\|\nabla f(\theta_t) - g_t\|^2,
\end{aligned}$$

where both the gradient bound and the assumption that $0 < a \leq f(\theta_t; \xi_t) + c = \tilde{F}_t^2$ are essentially used. Take an expectation to get

$$\mathbb{E}[\|\nabla F(\theta_t) - \frac{g_t}{2\tilde{F}_t}\|^2] \leq \frac{G_\infty^2}{2a^2}\mathbb{E}[|\tilde{F}_t - F(\theta_t)|^2] + \frac{1}{2a}\mathbb{E}[\|\nabla f(\theta_t) - g_t\|^2].$$

Similar to the proof for $(iv)$, we have

$$\mathbb{E}[|\tilde{F}_t - F(\theta_t)|^2] \leq \frac{1}{4a}\sigma_f^2.$$

This together with the variance assumption for $g_t$ gives

$$\mathbb{E}[\|\nabla F(\theta_t) - \frac{g_t}{2\tilde{F}_t}\|^2] \leq \frac{G_\infty^2}{8a^3}\sigma_f^2 + \frac{1}{2a}\sigma_g^2.$$

$\square$

**Lemma A.2.** *For any $T \geq 1$, we have*

*(i)* $\mathbb{E}\left[\sum_{t=1}^T v_t^\top r_{t+1} v_t\right] \leq \frac{nF(\theta_1)}{2\eta}$.

*(ii)* $\mathbb{E}\left[\sum_{t=1}^T m_{t-1}^\top r_{t+1} m_{t-1}\right] \leq \mathbb{E}\left[\sum_{t=1}^T m_t^\top r_{t+1} m_t\right] \leq \frac{2BnF(\theta_1)}{\eta}$.

*(iii)* $\mathbb{E}\left[\sum_{t=1}^T \|r_{t+1} m_t\|^2\right] \leq \frac{2BnF^2(\theta_1)}{\eta}$.

*(iv)* $\mathbb{E}\left[\sum_{t=1}^T g_t^\top r_{t+1} g_t\right] \leq \frac{8BnF(\theta_1)}{(1-\beta)^2\eta}$.

*(v)* $\mathbb{E}\left[\sum_{t=1}^T \|r_{t+1} g_t\|^2\right] \leq \frac{8BnF^2(\theta_1)}{(1-\beta)^2\eta}$.

*Proof.* From Algorithm 1 line 5, we have

$$r_{t,i} - r_{t+1,i} = 2\eta r_{t+1,i} v_{t,i}^2.$$

Taking summation over $t$ from 1 to $T$ gives

$$r_{1,i} - r_{T+1,i} = 2\eta \sum_{t=1}^T r_{t+1,i} v_{t,i}^2 \quad \Rightarrow \quad \sum_{t=1}^T r_{t+1,i} v_{t,i}^2 \leq \frac{r_{1,i}}{2\eta}.$$

From which we get

$$\sum_{t=1}^T v_t^\top r_{t+1} v_t = \sum_{i=1}^n \sum_{t=1}^T r_{t+1,i} v_{t,i}^2 \leq \frac{n\tilde{F}_1}{2\eta}.$$

Taking expectation and using (iii) in Lemma A.1 gives (i).

Recall that $m_t = 2(1 - \beta^t)\tilde{F}_t v_t$ and $\tilde{F}_t \leq \sqrt{B}$, we further get

$$\sum_{t=1}^T m_t^\top r_{t+1} m_t \leq 4B \sum_{t=1}^T v_t^\top r_{t+1} v_t = \frac{2Bn\tilde{F}_1}{\eta}.$$

Using $r_{t+1,i} \leq r_{t,i}$ and $m_{0,i} = 0$, we also have

$$\sum_{i=1}^n \sum_{t=1}^T r_{t+1,i} m_{t-1,i}^2 \leq \sum_{i=1}^n \sum_{t=1}^T r_{t,i} m_{t-1,i}^2 = \sum_{i=1}^n \sum_{t=1}^{T-1} r_{t+1,i} m_{t,i}^2 \leq \sum_{i=1}^n \sum_{t=1}^T r_{t+1,i} m_{t,i}^2. \quad (22)$$

Connecting the above two inequalities and taking expectation gives (ii).

Using $r_{t+1,i} \leq r_{1,i}$, the above inequality further implies

$$\sum_{t=1}^T \|r_{t+1} m_t\|^2 = \sum_{i=1}^n \sum_{t=1}^T r_{t+1,i}^2 m_{t,i}^2 \leq \sum_{i=1}^n \sum_{t=1}^T r_{1,i} r_{t+1,i} m_{t,i}^2$$

$$= \left(\sum_{i=1}^n \sum_{t=1}^T r_{t,i} m_{t,i}^2\right)\tilde{F}_1 \leq 2Bn\tilde{F}_1^2/\eta.$$

Taking expectation and using (ii) in Lemma A.1 gives (iii).

By $m_t = \beta m_{t-1} + (1 - \beta)g_t$, we have

$$\sum_{t=1}^{T} g_t^\top r_{t+1} g_t = \sum_{i=1}^{n} \sum_{t=1}^{T} r_{t+1,i} g_{t,i}^2 = \sum_{i=1}^{n} \sum_{t=1}^{T} r_{t+1,i} \left( \frac{1}{1-\beta} m_{t,i} - \frac{\beta}{1-\beta} m_{t-1,i} \right)^2$$

$$\leq \frac{2}{(1-\beta)^2} \sum_{i=1}^{n} \sum_{t=1}^{T} r_{t+1,i} m_{t,i}^2 + \frac{2\beta^2}{(1-\beta)^2} \sum_{i=1}^{n} \sum_{t=1}^{T} r_{t+1,i} m_{t-1,i}^2$$

$$\leq \frac{2(1+\beta^2)}{(1-\beta)^2} \sum_{t=1}^{T} m_t^\top r_{t+1} m_t \leq \frac{8Bn\tilde{F}_1}{(1-\beta)^2 \eta}.$$

Here the third inequality is by $(a+b)^2 \leq 2a^2 + 2b^2$; (22) and $0 < \beta < 1$ are used in the fourth inequality. Taking expectation and using (iii) in Lemma A.1 gives (iv).

Similar as the derivation for (ii), we have

$$\sum_{t=1}^{T} \|r_{t+1} g_t\|^2 \leq \left( \sum_{i=1}^{n} \sum_{t=1}^{T} r_{t,i} g_{t,i}^2 \right) \tilde{F}_1 \leq \frac{8Bn\tilde{F}_1^2}{(1-\beta)^2 \eta}.$$

Taking expectation and using (ii) in Lemma A.1 gives (v). $\qquad\square$

We are now ready to prove Theorem 4.2. The upper bound on $\sigma_g$ is given by (iv) in Lemma A.1. Since $f$ is $L$-smooth, we have

$$f(\theta_{t+1}) \leq f(\theta_t) + \nabla f(\theta_t)^\top (\theta_{t+1} - \theta_t) + \frac{L}{2} \|\theta_{t+1} - \theta_t\|^2. \tag{23}$$

Denoting $\eta_t = \eta/\tilde{F}_t$, the second term in the RHS of (23) can be expressed as

$$\nabla f(\theta_t)^\top (\theta_{t+1} - \theta_t)$$
$$= \nabla f(\theta_t)^\top (-2\eta r_{t+1} v_t)$$
$$= -\frac{1}{1-\beta^t} \nabla f(\theta_t)^\top \eta_t r_{t+1} m_t \quad (\text{since } m_t = 2(1-\beta^t)\tilde{F}_t v_t)$$
$$= -\frac{1}{1-\beta^t} \nabla f(\theta_t)^\top \eta_t r_{t+1} (\beta m_{t-1} + (1-\beta)g_t) \tag{24}$$
$$= -\frac{1-\beta}{1-\beta^t} \nabla f(\theta_t)^\top \eta_t r_{t+1} g_t - \frac{\beta}{1-\beta^t} \nabla f(\theta_t)^\top \eta_t r_{t+1} m_{t-1}$$
$$= -\frac{1-\beta}{1-\beta^t} \nabla f(\theta_t)^\top \eta_{t-1} r_t g_t + \frac{1-\beta}{1-\beta^t} \nabla f(\theta_t)^\top (\eta_{t-1} r_t - \eta_t r_{t+1}) g_t$$
$$\quad - \frac{\beta}{1-\beta^t} \nabla f(\theta_t)^\top \eta_t r_{t+1} m_{t-1}.$$

We further bound the second term and third term in the RHS of (24), respectively. For the second term, we note that $|\frac{1-\beta}{1-\beta^t}| \leq 1$ and

$$|\nabla f(\theta_t)^\top (\eta_{t-1} r_t - \eta_t r_{t+1}) g_t|$$
$$= |\nabla f(\theta_t)^\top \eta_{t-1} (r_t - r_{t+1}) g_t + \nabla f(\theta_t)^\top (\eta_{t-1} - \eta_t) r_{t+1} g_t|$$
$$= |\nabla f(\theta_t)^\top \eta_{t-1} (r_t - r_{t+1}) g_t + (\eta_{t-1} - \eta_t) g_t^\top r_{t+1} g_t$$
$$\quad + (\eta_{t-1} - \eta_t)(\nabla f(\theta_t) - g_t)^\top r_{t+1} g_t|$$
$$\leq \|\nabla f(\theta_t)\|_\infty |\eta_{t-1}| \|r_t - r_{t+1}\|_{1,1} \|g_t\|_\infty + |\eta_{t-1} - \eta_t| g_t^\top r_{t+1} g_t$$
$$\quad + |\eta_{t-1} - \eta_t| |(\nabla f(\theta_t) - g_t)^\top r_{t+1} g_t|$$
$$\leq (\eta G_\infty^2/\sqrt{a})(\|r_t\|_{1,1} - \|r_{t+1}\|_{1,1}) + (2\eta/\sqrt{a}) g_t^\top r_{t+1} g_t$$
$$\quad + (2\eta/\sqrt{a})|(\nabla f(\theta_t) - g_t)^\top r_{t+1} g_t|. \tag{25}$$

The third inequality holds because for a positive diagonal matrix $A$, $x^\top A y \leq \|x\|_\infty \|A\|_{1,1} \|y\|_\infty$, where $\|A\|_{1,1} = \sum_i a_{ii}$. The last inequality follows from the result $r_{t+1,i} \leq r_{t,i}$ for $i \in [n]$, the assumption $\|g_t\|_\infty \leq G_\infty$, $\tilde{F}_t \geq \sqrt{a}$, and (i) in Lemma (A.1).

For the third term in the RHS of (24), we note that

$$-\frac{\beta}{1 - \beta^t} \nabla f(\theta_t)^\top \eta_t r_{t+1} m_{t-1} \leq \frac{\beta\eta}{(1-\beta)\sqrt{a}} |\nabla f(\theta_t)^\top \eta_t r_{t+1} m_{t-1}|,$$

in which

$$\begin{aligned}
&|\nabla f(\theta_t)^\top r_{t+1} m_{t-1}| \\
&= |g_t^\top r_{t+1} m_{t-1} + (\nabla f(\theta_t) - g_t)^\top r_{t+1} m_{t-1}| \\
&\leq \frac{1}{2} g_t^\top r_{t+1} g_t + \frac{1}{2} m_{t-1}^\top r_{t+1} m_{t-1} + |(\nabla f(\theta_t) - g_t)^\top r_{t+1} m_{t-1}|,
\end{aligned} \tag{26}$$

where the last inequality is because for a positive diagonal matrix $A$, $x^\top A y \leq \frac{1}{2} x^\top A x + \frac{1}{2} y^\top A y$. Substituting (25) and (26) into (24), we get

$$\begin{aligned}
\nabla f(\theta_t)^\top (\theta_{t+1} - \theta_t) \leq{}& -\frac{1-\beta}{1-\beta^t} \nabla f(\theta_t)^\top \eta_{t-1} r_t g_t + \frac{\eta G_\infty^2}{\sqrt{a}} (\|r_t\|_{1,1} - \|r_{t+1}\|_{1,1}) \\
&+ \left(\frac{2\eta}{\sqrt{a}} + \frac{\beta\eta}{2(1-\beta)\sqrt{a}}\right) g_t^\top r_{t+1} g_t + \frac{\beta\eta}{2(1-\beta)\sqrt{a}} m_{t-1}^\top r_{t+1} m_{t-1} \\
&+ \frac{2\eta}{\sqrt{a}} |(\nabla f(\theta_t) - g_t)^\top r_{t+1} g_t| + \frac{\beta\eta}{(1-\beta)\sqrt{a}} |(\nabla f(\theta_t) - g_t)^\top r_{t+1} m_{t-1}|.
\end{aligned} \tag{27}$$

With (27), we take an conditional expectation on (23) with respect to $(\theta)$ and rearrange to get

$$\begin{aligned}
\frac{1-\beta}{1-\beta^t} \nabla f(\theta_t)^\top \eta_{t-1} r_t \nabla f(\theta_t) ={}& \mathbb{E}_{\xi_t} \left[ \frac{1-\beta}{1-\beta^t} \nabla f(\theta_t)^\top \eta_{t-1} r_t g_t \right] \\
\leq{}& \mathbb{E}_{\xi_t} \Bigg[ f(\theta_t) - f(\theta_{t+1}) + \frac{\eta G_\infty^2}{\sqrt{a}} (\|r_t\|_{1,1} - \|r_{t+1}\|_{1,1}) \\
&+ \left(\frac{2\eta}{\sqrt{a}} + \frac{\beta\eta}{2(1-\beta)\sqrt{a}}\right) g_t^\top r_{t+1} g_t + \frac{\beta\eta}{2(1-\beta)\sqrt{a}} m_{t-1}^\top r_{t+1} m_{t-1} \\
&+ \frac{2\eta}{\sqrt{a}} |(\nabla f(\theta_t) - g_t)^\top r_{t+1} g_t| \\
&+ \frac{\beta\eta}{(1-\beta)\sqrt{a}} |(\nabla f(\theta_t) - g_t)^\top r_{t+1} m_{t-1}| + \frac{L}{2} \|\theta_{t+1} - \theta_t\|^2 \Bigg],
\end{aligned} \tag{28}$$

where the assumption $\mathbb{E}_{\xi_t}[g_t] = \nabla f(\theta_t)$ is used in the first equality. Since $\xi_1, ..., \xi_t$ are independent random variables, we set $\mathbb{E} = \mathbb{E}_{\xi_1} \mathbb{E}_{\xi_2} ... \mathbb{E}_{\xi_T}$ and take a summation on (28) over $t$ from 1 to $T$ to get

$$\begin{aligned}
&\mathbb{E}\left[ \sum_{t=1}^T \frac{1-\beta}{1-\beta^t} \nabla f(\theta_t)^\top \eta_{t-1} r_t \nabla f(\theta_t) \right] \\
&\leq \mathbb{E}\Big[ f(\theta_1) - f(\theta_{T+1}) \Big] + \frac{\eta G_\infty^2}{\sqrt{a}} \mathbb{E}\Big[ \|r_1\|_{1,1} - \|r_{T+1}\|_{1,1} \Big] \\
&+ \left(\frac{2\eta}{\sqrt{a}} + \frac{\beta\eta}{2(1-\beta)\sqrt{a}}\right) \mathbb{E}\left[ \sum_{t=1}^T g_t^\top r_{t+1} g_t \right] + \frac{\beta\eta}{2(1-\beta)\sqrt{a}} \mathbb{E}\left[ \sum_{t=1}^T m_{t-1}^\top r_t m_{t-1} \right] \\
&+ \frac{2\eta}{\sqrt{a}} \mathbb{E}\left[ \sum_{t=1}^T |(\nabla f(\theta_t) - g_t)^\top r_{t+1} g_t| \right] \\
&+ \frac{\beta\eta}{(1-\beta)\sqrt{a}} \mathbb{E}\left[ \sum_{t=1}^T |(\nabla f(\theta_t) - g_t)^\top r_{t+1} m_{t-1}| \right] + \frac{L}{2} \mathbb{E}\left[ \sum_{t=1}^T \|\theta_{t+1} - \theta_t\|^2 \right].
\end{aligned} \tag{29}$$

Below we bound each term in (29) separately. By the Cauchy-Schwarz inequality, we get

$$
\begin{aligned}
&\mathbb{E}\left[\sum_{t=1}^{T}|(\nabla f(\theta_t) - g_t)^{\top} r_{t+1} m_{t-1}|\right] \\
&\leq \mathbb{E}\left[\sum_{t=1}^{T}\|\nabla f(\theta_t) - g_t\|\|r_{t+1} m_{t-1}\|\right] \\
&\leq \mathbb{E}\left[\left(\sum_{t=1}^{T}\|\nabla f(\theta_t) - g_t\|^2\right)^{1/2}\left(\sum_{t=1}^{T}\|r_{t+1} m_{t-1}\|^2\right)^{1/2}\right] \\
&\leq \left(\mathbb{E}\left[\sum_{t=1}^{T}\|\nabla f(\theta_t) - g_t\|^2\right]\right)^{1/2}\left(\mathbb{E}\left[\sum_{t=1}^{T}\|r_{t+1} m_{t-1}\|^2\right]\right)^{1/2} \\
&\leq \sqrt{2BnT/\eta}F(\theta_1)\sigma_g,
\end{aligned}
\tag{30}
$$

where Lemma A.1 (ii) and the bounded variance assumption were used. We replace $m_{t-1}$ in (30) by $g_t$ and use Lemma A.1 (v) to get

$$
\begin{aligned}
&\mathbb{E}\left[\sum_{t=1}^{T}|(\nabla f(\theta_t) - g_t)^{\top} r_{t+1} g_t|\right] \\
&\leq \left(\mathbb{E}\left[\sum_{t=1}^{T}\|\nabla f(\theta_t) - g_t\|^2\right]\right)^{1/2}\left(\mathbb{E}\left[\sum_{t=1}^{T}\|r_{t+1} g_t\|^2\right]\right)^{1/2} \\
&\leq \frac{2\sqrt{2BnT/\eta}F(\theta_1)\sigma_g}{1-\beta}.
\end{aligned}
\tag{31}
$$

By (12), the last term in (29) is bounded above by

$$
\frac{L}{2}\mathbb{E}\left[\sum_{t=0}^{\infty}\|\theta_{t+1} - \theta_t\|^2\right] \leq \frac{L\eta n}{2}F^2(\theta_1).
\tag{32}
$$

Substituting Lemma A.1 (i) (iii), (32), (31), (30) into (29) to get

$$
\begin{aligned}
\mathbb{E}\left[\sum_{t=1}^{T}\frac{1-\beta}{1-\beta^t}\nabla f(\theta_t)^{\top}\eta_{t-1} r_t \nabla f(\theta_t)\right] \leq\ &(f(\theta_1) - f^*) + \frac{\eta G_\infty^2}{\sqrt{a}}nF(\theta_1) \\
&+ \left(\frac{2}{\sqrt{a}} + \frac{\beta}{2(1-\beta)\sqrt{a}}\right)\frac{8BnF(\theta_1)}{(1-\beta)^2} + \frac{\beta BnF(\theta_1)}{(1-\beta)\sqrt{a}} \\
&+ \frac{(4+\beta)\sqrt{2B\eta}}{(1-\beta)\sqrt{a}}F(\theta_1)\sqrt{nT}\sigma_g + \frac{L\eta n}{2}F^2(\theta_1).
\end{aligned}
\tag{33}
$$

Note that the left hand side is bounded from below by

$$
(1-\beta)\frac{\eta}{\sqrt{B}}\mathbb{E}\left[\min_i r_{T,i}\sum_{t=1}^{T}\|\nabla f(\theta_t)\|^2\right],
$$

where we used $|\frac{1-\beta}{1-\beta^t}| \geq 1-\beta$ and $\eta_t \geq \eta/\sqrt{B}$. Thus we have

$$
\mathbb{E}\left[\min_i r_{T,i}\sum_{t=1}^{T}\|\nabla f(\theta_t)\|^2\right] \leq \frac{C_1 + C_2 n + C_3 \sigma_g \sqrt{nT}}{\eta},
$$

where

$$C_1 = \frac{(f(\theta_1) - f^*)\sqrt{B}}{1 - \beta},$$

$$C_2 = \frac{\sqrt{B}\eta G_\infty^2 F(\theta_1)}{(1-\beta)\sqrt{a}} + \left(\frac{2}{\sqrt{a}} + \frac{\beta}{2(1-\beta)\sqrt{a}}\right)\frac{8B^{3/2}F(\theta_1)}{(1-\beta)^3}$$

$$+ \frac{\beta B^{3/2}F(\theta_1)}{(1-\beta)^2\sqrt{a}} + \frac{\sqrt{B}L\eta}{2(1-\beta)^2}F^2(\theta_1),$$

$$C_3 = \frac{(4+\beta)B\sqrt{2\eta}}{(1-\beta)\sqrt{a}}F(\theta_1).$$

## A.2 PROOF OF THEOREM 4.3

First note that by (iv) in Lemma A.1, $\max\{\sigma_g, \sigma_f\} \leq \max\{G_\infty, B\}$.

Recall that $F(\theta) = \sqrt{f(\theta) + c}$, then for any $x, y \in \{\theta_t\}_{t=0}^T$ we have

$$\|\nabla F(x) - \nabla F(y)\| = \left\|\frac{\nabla f(x)}{2F(x)} - \frac{\nabla f(y)}{2F(y)}\right\|$$

$$= \frac{1}{2}\left\|\frac{\nabla f(x)(F(y) - F(x))}{F(x)F(y)} + \frac{\nabla f(x) - \nabla f(y)}{F(y)}\right\|$$

$$\leq \frac{G_\infty}{2(F(\theta^*))^2}|F(y) - F(x)| + \frac{1}{2F(\theta^*)}\|\nabla f(x) - \nabla f(y)\|.$$

One may check that

$$|F(y) - F(x)| \leq \frac{G_\infty}{2F(\theta^*)}\|x - y\|.$$

These together with the $L$-smoothness of $f$ lead to

$$\|\nabla F(x) - \nabla F(y)\| \leq L_F\|x - y\|,$$

where

$$L_F = \frac{1}{2\sqrt{f(\theta^*) + c}}\left(L + \frac{G_\infty^2}{2(f(\theta^*) + c)}\right).$$

This confirms the $L_F$-smoothness of $F$, which yields

$$F(\theta_{t+1}) - F(\theta_t) \leq \nabla F(\theta_t)^\top(\theta_{t+1} - \theta_t) + \frac{L_F}{2}\|\theta_{t+1} - \theta_t\|^2$$

$$= (\nabla F(\theta_t) - \frac{g_t}{2\tilde{F}_t})^\top(\theta_{t+1} - \theta_t) + (\frac{g_t}{2\tilde{F}_t} - \frac{1-\beta^t}{1-\beta}v_t)^\top(\theta_{t+1} - \theta_t)$$

$$+ (\frac{1-\beta^t}{1-\beta}v_t)^\top(\theta_{t+1} - \theta_t) + \frac{L_F}{2}\|\theta_{t+1} - \theta_t\|^2.$$

Summation of the above over $t$ from 1 to $T$ and taken with the expectation gives

$$\mathbb{E}[F(\theta_{T+1}) - F(\theta_1)] \leq \sum_{i=1}^4 S_i, \tag{34}$$

where

$$S_1 = \mathbb{E}\left[\sum_{t=1}^{T} \frac{1-\beta^t}{1-\beta} v_t^\top (\theta_{t+1} - \theta_t)\right],$$

$$S_2 = \mathbb{E}\left[\sum_{t=1}^{T} (\frac{g_t}{2\tilde{F}_t} - \frac{1-\beta^t}{1-\beta} v_t)^\top (\theta_{t+1} - \theta_t)\right],$$

$$S_3 = \mathbb{E}\left[\sum_{t=1}^{T} (\nabla F(\theta_t) - \frac{g_t}{2\tilde{F}_t})^\top (\theta_{t+1} - \theta_t)\right],$$

$$S_4 = \mathbb{E}\left[\sum_{t=1}^{T} \frac{L_F}{2} \|\theta_{t+1} - \theta_t\|^2\right].$$

Below we bound $S_1, S_2, S_3, S_4$ separately. To bound $S_1$, we first note that

$$r_{t+1,i} - r_{t,i} = -2\eta r_{t+1,i} v_{t,i}^2 = v_{t,i}(-2\eta r_{t+1,i} v_{t,i}) = v_{t,i}(\theta_{t+1,i} - \theta_i)$$

from which we get

$$\begin{aligned}
S_1 &= \mathbb{E}\left[\sum_{t=1}^{T} \frac{1-\beta^t}{1-\beta} v_t^\top (\theta_{t+1} - \theta_t)\right] \\
&= \mathbb{E}\left[\sum_{i=1}^{n}\sum_{t=1}^{T} \frac{1-\beta^t}{1-\beta} (r_{t+1,i} - r_{t,i})\right] \\
&\leq \mathbb{E}\left[\sum_{i=1}^{n}\sum_{t=1}^{T} r_{t+1,i} - r_{t,i}\right] \quad \text{(Since } r_{t+1,i} \leq r_{t,i}) \\
&= \sum_{i=1}^{n} \mathbb{E}[r_{T+1,i}] - n\mathbb{E}[\tilde{F}_1].
\end{aligned}$$

For $S_2$, we have

$$\begin{aligned}
S_2 &= \mathbb{E}\left[\sum_{t=1}^{T} (\frac{g_t}{2\tilde{F}_t} - \frac{1-\beta^t}{1-\beta} v_t)^\top (\theta_{t+1} - \theta_t)\right] \\
&= \mathbb{E}\left[\sum_{i=1}^{n}\sum_{t=1}^{T} (-\frac{1}{2\tilde{F}_t}\frac{\beta}{1-\beta} m_{t-1,i})^\top (2\eta r_{t+1,i} v_{t,i})\right] \\
&\leq \frac{\beta\eta}{(1-\beta)\sqrt{a}} \mathbb{E}\left[\sum_{i=1}^{n}\sum_{t=1}^{T} r_{t+1,i} m_{t-1,i} v_{t,i}\right] \\
&\leq \frac{\beta\eta}{(1-\beta)\sqrt{a}} \mathbb{E}\left[\left(\sum_{i=1}^{n}\sum_{t=1}^{T} r_{t+1,i} m_{t-1,i}^2\right)^{1/2}\left(\sum_{i=1}^{n}\sum_{t=1}^{T} r_{t+1,i} v_{t,i}^2\right)^{1/2}\right] \\
&\leq \frac{\beta\sqrt{B}nF(\theta_1)}{(1-\beta)\sqrt{a}},
\end{aligned}$$

where the fourth inequality is by the Cauchy-Schwarz inequality, the last inequality is by Lemma A.1 (i) (ii).

For $S_3$, by the Cauchy-Schwarz inequality, we have

$$
\begin{aligned}
S_3 &= \mathbb{E}\left[\sum_{t=1}^{T}(\nabla F(\theta_t) - \frac{g_t}{2\tilde{F}_t})^{\top}(\theta_{t+1} - \theta_t)\right] \\
&\leq \mathbb{E}\left[\sum_{t=1}^{T}\|\nabla F(\theta_t) - \frac{g_t}{2\tilde{F}_t})\|\|\theta_{t+1} - \theta_t)\|\right] \\
&\leq \mathbb{E}\left[\left(\sum_{t=1}^{T}\|\nabla F(\theta_t) - \frac{g_t}{2\tilde{F}_t})\|^2\right)^{1/2}\left(\sum_{t=1}^{T}\|\theta_{t+1} - \theta_t\|^2\right)^{1/2}\right] \\
&\leq \left(\mathbb{E}\left[\sum_{t=1}^{T}\|\nabla F(\theta_t) - \frac{g_t}{2\tilde{F}_t})\|^2\right]\right)^{1/2}\left(\mathbb{E}\left[\sum_{t=1}^{T}\|\theta_{t+1} - \theta_t\|^2\right]\right)^{1/2} \\
&\leq F(\theta_1)\sqrt{\eta n T}\sqrt{\frac{G_{\infty}^2}{8a^3}\sigma_f^2 + \frac{1}{2a}\sigma_g^2},
\end{aligned}
$$

where the last inequality is by (vi) in Lemma A.1 and (12) in Theorem 4.1.

For $S_4$, also by (12) in Theorem 4.1, we have

$$
S_4 = \frac{L_F}{2}\mathbb{E}\left[\sum_{t=1}^{T}\|\theta_{t+1} - \theta_t\|^2\right] \leq \frac{L_F\eta n F^2(\theta_1)}{2}.
$$

With the above bounds on $S_1, S_2, S_3, S_4$, (34) can be rearranged as

$$
\begin{aligned}
&F(\theta^*) - \frac{\beta\sqrt{B}nF(\theta_1)}{(1-\beta)\sqrt{a}} - F(\theta_1)\sqrt{\eta n T}\sqrt{\frac{G_{\infty}^2}{4a^3}\sigma_f^2 + \frac{1}{a}\sigma_g^2} - \frac{L_F\eta n F^2(\theta_1)}{2} \\
&\leq \sum_{i=1}^{n}\mathbb{E}[r_{T+1,i}] - n\mathbb{E}[\tilde{F}_1] + F(\theta_1) \\
&\leq \left(\min_{i}\mathbb{E}[r_{T+1,i}] + (n-1)\mathbb{E}[\tilde{F}_1]\right) - (n-1)\mathbb{E}[\tilde{F}_1] + \left(F(\theta_1) - \mathbb{E}[\tilde{F}_1]\right) \\
&\leq \min_{i}\mathbb{E}[r_{T+1,i}] + \mathbb{E}[|F(\theta_1) - \tilde{F}_1|] \\
&\leq \min_{i}\mathbb{E}[r_{T+1,i}] + \frac{1}{2\sqrt{a}}\sigma_f,
\end{aligned}
$$

where (iii) in Lemma A.1 was used. Hence,

$$
\min_{i}\mathbb{E}[r_{T,i}] \geq \max\{F(\theta^*) - \eta D_1 - \beta D_2 - \sigma D_3, 0\},
$$

where $\sigma = \max\{\sigma_f, \sigma_g\}$ and

$$
D_1 = \frac{L_F n F^2(\theta_1)}{2}, \quad D_2 = \frac{\sqrt{B}nF(\theta_1)}{(1-\beta)\sqrt{a}},
$$

$$
D_3 = \frac{1}{2\sqrt{a}} + F(\theta_1)\sqrt{\eta n T}\sqrt{\frac{G_{\infty}^2}{4a^3} + \frac{1}{a}}.
$$

## A.3 PROOF OF THEOREM 4.4

Using the same argument as for (iv) in Lemma A.2, we have

$$
\sum_{i=1}^{n}\sum_{t=1}^{T}r_{t+1,i}g_{t,i}^2 \leq \frac{8Bn\sqrt{f_1(\theta_1) + c}}{(1-\beta)^2\eta}.
$$

With this estimate and the convexity of $f_t$, the regret can be bounded by

$$
\begin{aligned}
R(T) = \sum_{t=1}^{T} f_t(\theta_t) - f_t(\theta^*) &\leq \sum_{t=1}^{T} g_t^\top (\theta_t - \theta^*) \\
&\leq \sum_{i=1}^{n} \sum_{t=1}^{T} |g_{t,i}| \sqrt{r_{t+1,i}} \frac{|\theta_{t,i} - \theta_i^*|}{\sqrt{r_{t+1,i}}} \\
&\leq \left( \sum_{i=1}^{n} \sum_{t=1}^{T} r_{t+1,i} g_{t,i}^2 \right)^{1/2} \left( \sum_{i=1}^{n} \sum_{t=1}^{T} \frac{|\theta_{t,i} - \theta_i^*|^2}{r_{t+1,i}} \right)^{1/2} \\
&\leq \frac{2 D_\infty \sqrt{2B}}{1 - \beta} (f_1(\theta_1) + c)^{1/4} \sqrt{nT/\eta} \left( \sum_{i=1}^{n} \frac{1}{r_{T+1,i}} \right)^{1/2},
\end{aligned}
$$

where the fourth inequality is by the Cauchy-Schwarz inequality, and the assumption $\|x - y\|_\infty \leq D_\infty$ for all $x, y \in \mathcal{F}$ is used in the last inequality.

