# OpenReview forum: "SGDEM: stochastic gradient descent with energy and momentum"
_ICLR.cc/2022/Conference — ICLR 2022 Submitted_

### Official Review · Reviewer_64qc · 2021-11-01

**Correctness:** 4
**Technical Novelty And Significance:** 2
**Empirical Novelty And Significance:** 2
**Recommendation:** 6
**Confidence:** 5

**Main Review:**

The main result of the paper is combining the two existing algorithms: AEGM and momentum method. However, due to the momentum update of the gradient estimator, the analysis is different from the existing analysis of the AEGM and is more complex to some degree.  However, there are some issues w.r.t. the current analysis, for which the reviewer has no idea how one can fix that.

The experiments of the paper have shown some performance improvements.

Overall, the reviewer thinks this work is marginally below the acceptance level.

**Summary Of The Paper:**

In this paper, the authors propose an algorithm called SGDEM, which is a combination of the AEGD method and the momentum method. For the proposed method, the authors have shown a convergence to the stationary point for the nonconvex case and a regret bound for the online convex case. Numerical experiments are done for several neural network training problems.



**Summary Of The Review:**

The main issue:

In Theorem 4.2, the reviewer finds the LHS of the bound is deterministic (after taking expectation), while the RHS is stochastic due to the presence of $\min_i r_{T,i}$, which is confusing to the reviewer.

This seems to be the following mistake in the analysis. In the first equation under Eq. (33), the authors seems to argue that
$$E\Big[ \min_i r_{T,i}\cdot \sum_{t=0}^{T-1}||\nabla f(\theta_t)||^2 \Big] \geq \min_i r_{T,i}\cdot E\Big[ \sum_{t=0}^{T-1}||\nabla f(\theta_t)||^2 \Big].$$
However, this is not right in general since $\min_i r_{T,i}$ is a random variable and is dependent on $E\Big[\sum_{t=0}^{T-1}||\nabla f(\theta_t)||^2 \Big]$. It is also not possible to write $E\Big[ \min_i r_{T,i}\cdot \sum_{t=0}^{T-1}||\nabla f(\theta_t)||^2 \Big] \geq E\Big[\min_i r_{T,i}\Big]\cdot E\Big[ \sum_{t=0}^{T-1}||\nabla f(\theta_t)||^2 \Big].$

The reviewer does not have any idea to fix this issue. And it should be formally pointed out by the authors. In the worst case, the authors should at least formally make the following assumption, and point out that this assumption is UNCHECKABLE.

Assumption: there exists a constant W>0 s.t. $\min_i r_{T,i}\geq W$ for any $T$ and any iteration sequence generated by the algorithm.







Minor issues:

1. The authors use both $f(\theta;\xi)$ and $f(\theta,\xi)$ in this paper, please unify the notation.

2. On page 2, related works, the "differentialequations" should be "differential equations".

3. In the first equation under (13), a coefficient of $\frac{1}{1-\beta^t}$ is missing.

4. In the second equation under (13), the "$\leq$" should be "$\geq$" and "\sqrt{a}" should be "$\sqrt{B}$".

5. Throughout the paper, the authors use the notation of $w^Tuv$. The authors should formally write $w^T(u\odot v)$, in order to distinguish from the common understanding $(w^Tu)\cdot v$.


--------------------------------------------------------------------------------------------------------
After revision, the author clear the mistake mentioned above. I have changed the score from 5 to 6,

---

> ### Author Response · Authors · 2021-11-20
> **Response to Reviewer 64qc**
>
> For the main issue: Thank you for pointing this error out!  We have restarted the results, equivalent to adding the assumption as you suggested.
>
> For minor issues: Thank you for your careful reading! 1-4 have been fixed. For 5, we have made the notation clearer in Remark 3.1 and on page 5.

---

> > ### Comment · Reviewer_64qc · 2021-11-25
> > **Score adjusted**
> >
> > Thanks for the revision. The restated results are correct now.
> > However, the correct one is still somehow blur on the convergence rate, (actually, the assumption I talked about is uncheckable). This might be the internal drawback of this approach. So, I may only adjust the score to be 6,

---

### Official Review · Reviewer_Su6j · 2021-11-02

**Correctness:** 3
**Technical Novelty And Significance:** 2
**Empirical Novelty And Significance:** 1
**Recommendation:** 5
**Confidence:** 4

**Details Of Ethics Concerns:**

NA, optimization methods.

**Main Review:**

Strengths:

1: This paper proposes an improved variant of AEGD by incorporating the energy and momentum at the same time.

2: Rigorous theoretical analysis is provided to verify the effectiveness of the proposed algorithm.

3: This paper is well-organized and easy to read. The background knowledge is presented well and the related papers are cited. The readers can easily understand the paper.

Weaknesses:

1: The contribution of SGDEM over AEGD is limited. Although theoretical analysis is provided to verify the effectiveness of the proposed algorithm, the advantages of SGDEM over the AEGD are unclear.  As an improved version of AEGD, I believe detailed comparisons of theoretical results between these two methods are required.

2:  The motivation to incorporate the momentum mechanism is straightforward since the momentum is widely used for optimization methods such as the popular SGD with momentum. However, the relation between the energy and the momentum is unclear. If it is just a combination of the known energy method in AEGD and the momentum method in SGD with momentum, the idea is not well-motivated.

3: The experimental results are weak. For most of the experiments, the proposed method performs worse than baseline AEGD even with the existence of oscillation.

4: The experiments are only conducted for vision tasks while NLP is a very important application in deep learning. The optimization method should also be essentially tested for NLP tasks.

**Summary Of The Paper:**

This paper studies stochastic gradient descent methods for general non-convex stochastic optimization problems. The proposed method SGDEM is an improved version of AEGD by incorporating both energy and momentum at the same time, which can achieve unconditional energy stability property. The authors provide an energy-dependent convergence rate for the non-convex stochastic setting and a regret bund in the online setting.

**Summary Of The Review:**

I have checked the response. I still think the idea is not novel enough and the experimental results are weak. Therefore, I would like to keep my score.

---

> ### Author Response · Authors · 2021-11-20
> **Response to Reviewer Su6j**
>
> Thanks for your questions and suggestions!
>
> 1. Thank you for the suggestion. We have added a brief comparison of theoretical results on page 2.
>
> 2. Thank you for pointing this out. The relation between the energy and the momentum in the algorithm is realized through relating $m_t$ ( as an approximation to $\nabla f$)  to $v_t$ (as an approximation of $\nabla F = \frac{\nabla f}{2\sqrt{f+c}}$), where $v_t$ is used to update the energy $r_{t}$. Also, a key feature of the proposed algorithm is that it incorporates momentum into AEGD without changing the overall structure of the AEGD algorithm (the update of $r$ and $\theta$ remains the same) so that the proposed algorithm still enjoys the unconditional energy stability property (Theorem 4.1).
>
> 3. In our work, the improvement of the proposed algorithm over AEGD is shown mainly in terms of the convergence speed (which is realized through reducing the oscillations) rather than the generalization performance. Even when we compare the generalization performance, the proposed algorithm outperforms AEGD in examples such as VGG-16, DenseNet-121 on CIFAR-100.
>
> 4. Thanks for the suggestion. We have attempted NLP tasks as suggested, but the result is not that good. We plan to conduct more fined experiments on NLP in future work.

---

### Official Review · Reviewer_e7GR · 2021-11-02

**Correctness:** 4
**Technical Novelty And Significance:** 2
**Empirical Novelty And Significance:** 2
**Recommendation:** 5
**Confidence:** 4

**Main Review:**

Strengths:
- Convergence analysis for SGDEM are shown for both nonconvex and convex settings. I have checked the proofs and find no problem but there is a chance I miss something.
- Numerical experiments on different deep learning models on two CIFAR datasets illustate the advantage of SGDEM over other baselines.

Weaknesses:
- There is not much novelty in algorithm design as the idea of using momentum is quite common.
- For nonconvex analysis, the assumption that the gradient and function value are bounded above is quite strong to me. It limits the applicability of the proposed method.
- For convex analysis, the assumption that the domain is bounded and bounded function value are also strong.
- Numerical experiments only consider deep learning examples which are nonconvex objectives. I wonder how SGDEM performs in objectives other than neural network. Adding examples including convex objectives is also recommended.

**Summary Of The Paper:**

The paper  extends the adaptive gradient descent with energy method by incorporating momentum to obtain a new variant called stochastic gradient descent with energy and momentum (SGDEM). Convergence analysis is provided for nonconvex and convex expectation problems. Numerical experiments demonstrate the effectiveness of SGDEM over AEGD and other baselines.

**Summary Of The Review:**

The paper contains substantial novelty in providing convergence analysis for SGDEM. However, the theoretical results rely on additional assumptions which are rather strong including bounded gradient, bounded function value (and bounded domain for convex setting) apart from standard assumptions. Numerical experiments only consider deep learning examples which is not extensive enough to illustrate the practical performance of SGDEM.

---

> ### Author Response · Authors · 2021-11-20
> **Response to Reviewer e7GR**
>
> Thank you for your comments.
>
> 1. Though the idea of momentum has been long known, the novelty here is about how to integrate the energy with momentum, among different options. A key feature of the proposed algorithm is that it incorporates momentum into AEGD without changing the overall structure of the AEGD algorithm (the update of $r$ and $\theta$ remain the same) so that the proposed algorithm still enjoys the unconditional energy stability property (Theorem 4.1).
>
> 2. The bounded stochastic gradient assumption is standard in the nonconvex stochastic analysis [Bottou18]. As for the upper bound on the stochastic function value, we have added an explanation in Remark 4.2.
>
> 3. The bounded domain is typically enforced by projection onto the feasible set $\mathcal{F}$, which is a standard assumption used in online convex programming [Zin03]. As for the upper bound on the function value, we have added an explanation in Remark 4.4.
>
> 4. Thank you for the suggestion. The proposed algorithm is aimed at solving stochastic optimization problems, among which training neural networks is of the main interest in current research. Thus we mainly consider neural network examples. As for convex objectives, we conducted experiments about training Logistic regression on the MNIST dataset, which also verifies that the proposed algorithm improves the performance of AEGD, especially in training speed. We didn't include the result in our paper because the difference between the performance of each method is not as obvious as that in CIFAR examples, and we believe that the results in CIFAR and ImageNet examples are good enough to show the good performance of the proposed algorithm.
>
> [Bottou18] Leon Bottou, Frank E. Curtis, and Jorge Nocedal.  Optimization methods for large-scale machine learning. SIAM Rev., 60(2):223–311, 2018.
>
> [Zin03] Martin Zinkevich. Online convex programming and generalized infinitesimal gradient ascent. In Proceedings of the Twentieth International Conference on International Conference on Machine Learning, ICML, pp. 928–935, 2003.

---

> > ### Comment · Reviewer_e7GR · 2021-11-27
> > **Response Feedback**
> >
> > Thank you for clarifying my concerns. Regarding the bounded gradient assumption, I believe this assumption is commonly used to analyze the convergence of adaptive stochastic gradient methods while non-adaptive methods such as SGD or variance reduced methods like SVRG, SARAH, STORM do not need this assumption. The rate of convergence presented does not improve the rate of standard SGD which does not need the bounded gradient assumption.
> >
> > After reading your response and discussion from other reviewers, I decide to still keep my score.

---

### Official Review · Reviewer_TrUv · 2021-11-03

**Correctness:** 2
**Technical Novelty And Significance:** 2
**Empirical Novelty And Significance:** 2
**Recommendation:** 5
**Confidence:** 5

**Main Review:**

The strengths are:
- The authors propose a new algorithm based on the idea of 'energy' variable from the work  [AEGD: Adaptive Gradient Descent with Energy].
- They prove the unconditional energy stability property for SGDEM that is similar to AEGD and does not depend on the transformed momentum term $v_t$. They provide some convergence bounds for SGDEM for general stochastic nonconvex setting, and a regret bound for the online convex framework.
- Numerical results are shown for their method and other state-of-the-art methods for training neural networks on ImageNet and CIFAR dataset.

The weaknesses are:
- Firstly, the theoretical results use a lot of assumptions. Assumption 4.1 involved both the first and zero-th oracles, while Theorem 4.2 still needs that the gradient and function value are bounded. (Why assuming bounded variance while the gradients are bounded?). Similarly, Theorem 4.3 assumes that all iterates and the function value are bounded. The bounded function value is a strong assumption because it assumes that the algorithm can not go to some bad points with large output. We also need that $a$ is strictly larger than 0.
- Secondly, the convergence of this method depends heavily on the lower bound of $r_{T,i}$. We know that the expectation of this sequence is strictly decreasing, and this term always appears at the denominator of the bound. Hence if this term goes to 0, the results are not informative. The authors can not prove a lower bound for $r_{T,i}$ (away from 0) even in Theorem 4.3, since the right-hand side of equation (15) can be 0 especially when $F(\theta^*) -\beta D_2 < 0$, no matter how they choose the learning rate. In addition, we can not choose an arbitrarily small learning rate to satisfy a positive lower bound, because it may harm the result of Theorem 4.2.
- Finally, the experiment of SGDEM does not show encouraging performance (over existing methods) for CIFAR datasets.

Other comments:
- The name SGDEM appeared in another work (https://arxiv.org/pdf/2102.13653v1.pdf) which was published in February 2021. If this work was published later than that time period, I strongly suggest that the authors change the abbreviated name of your method to avoid confusion, and to distinguish the two methods.
- In Algorithm 1, it is not quite reasonable to state a requirement in the beginning that the (stochastic) function value of the algorithm at $t$ is larger than minus $c$, for all iteration $t$. Since this is a hyper-parameter, it must be chosen in advance to satisfy that $-c$ is a lower bound of every possible input. Note that this hyper parameter $c$ also depends on the stochastic function value, not only $f$

**Summary Of The Paper:**

In this paper, the authors propose a new algorithm called Stochastic Gradient Descent with Energy and Momentum (SGDEM) for non-convex stochastic optimization problems. The idea of 'energy' variable is from the work  [AEGD: Adaptive Gradient Descent with Energy]. The authors prove some similar property for SGDEM (unconditional energy stability) that does not depend on the transformed momentum term $v_t$. They prove some convergence bounds for SGDEM for general stochastic nonconvex setting, and a regret bound for the
online convex framework. In addition, they provide numerical results for their method in comparison with others for training neural networks.

**Summary Of The Review:**

To summarize, the authors proposed a momentum method based on AEGD using gradient and function value information. However, the theoretical and experimental results are not convincing enough. Hence I will encourage the authors to improve this paper and resubmit to another venue.

---
I thank the authors for your responses and the improvement of your manuscript. After the discussion period, I decided to keep my initial suggestion.

---

> ### Author Response · Authors · 2021-11-20
> **Response to Reviewer TrUv**
>
> Thanks for your questions and suggestions. Below we respond to your comments point by point.
>
> For the weaknesses you are concerned about:
>
> 1. Thanks for pointing this out. In Theorem 4.2, the assumption that both the magnitude of the stochastic gradient and the variance are bounded is now reduced to only assuming the magnitude of the stochastic gradient is bounded, which is standard in the stochastic analysis [Bottou18]. The same for the assumption on the stochastic function value.
> In Theorem 4.4, the bounded domain is typically enforced by projection onto the feasible set $\mathcal{F}$, which is a standard assumption used in online convex programming [Zin03]. As for the bound on the stochastic function value, we added some explanations on page 4 and in Remark 4.2.
>
> 2. Indeed, this is the main theoretical challenge for the present algorithm. The sufficient condition provided only indicates the possibility of achieving a lower bound for $r_{T, i}$, it is far from an optimal estimate.  While as evidenced by our experiments, $r_{T,i}$ stays far away from $0$ for a large range of choices of $\eta$.
>
> 3. The improvement of the proposed algorithm over AEGD is shown mainly in terms of the convergence speed (which is realized through reducing the oscillations) rather than the generalization performance. Even when we compare the generalization performance, the proposed algorithm outperforms AEGD in examples such as VGG-16, DenseNet-121 on CIFAR-100. Compared with other methods, the proposed algorithm shows better generalization performance in all CIFAR examples.
>
> For your other comments:
>
> 1. Thank you for this information. We have changed the short name to SGEM (Stochastic Gradient with Energy and Momentum).
>
> 2. Thanks for pointing out this problem. We have added an explanation of why $c$ can be easily chosen on page 4.
>
> [Bottou18] Leon Bottou, Frank E. Curtis, and Jorge Nocedal.  Optimization methods for large-scale machine learning.SIAM Rev., 60(2):223–311, 2018.
>
> [Zin03] Martin Zinkevich.  Online convex programming and generalized infinitesimal gradient ascent.  InProceedings of the Twentieth International Conference on International Conference on MachineLearning, ICML, pp. 928–935, 2003.

---

### Decision · Program_Chairs · 2022-01-20

**Decision:**

Reject

**Comment:**

The reviewers have the following concerns:
1. The theoretical results for the proposed method are weak. Theorem 4.2 cannot be considered as a convergence result, because the bound depends on some random variables $r_{T,i}$. The reviewers agree that a proper analysis would require some knowledge on the lower bound of these variables. Although there is some empirical explanation for this, the lower bounded assumption of  $r_{T,i}$ is not theoretically justified. The authors acknowledge that this is the main challenge for the present algorithm. In addition, the analysis requires bounded gradient and bounded function value, which is also strong for nonconvex settings.
2. The empirical performance is not strong. In most experiments, the proposed method is not better than the baseline AEGD. The novelty and contribution of SGEM over AEGD is quite limited, since the idea of adding momentum is not new.

The suggestions to improve this paper are as follows
1. Since the lower bounded assumption on $r_{T,i}$ is not standard and hard to verify, the authors might consider analyzing a theoretical guarantee for it. On the other hand, they could verify more experiments with various data sets to have some sense whether this assumption may be true or not. Next, please try to relax the strong assumptions as discussed.
2. It is better if the authors can show the performance of SGEM for convex settings, and for other deep learning tasks (e.g. NLP) as suggested by the reviewers.

The authors should consider to improve the paper based on the reviewers' comments and suggestions and resubmit this paper in the future venues.